# Carbazole Derivatives as STAT Inhibitors: An Overview

**Anna Caruso** [1,†], **Alexia Barbarossa** [2,†], **Alessia Carocci** [2], **Giovanni Salzano** [3], **Maria Stefania Sinicropi** [1,*] and **Carmela Saturnino** [3]

1   Department of Pharmacy, Health and Nutritional Sciences, University of Calabria, 87036 Arcavacata di Rende, Italy; anna.caruso@unical.it
2   Department of Pharmacy-Pharmaceutical Sciences, University of Bari Aldo Moro, 70126 Bari, Italy; alexia.barbarossa@uniba.it (A.B.); alessia.carocci@uniba.it (A.C.)
3   Department of Science, University of Basilicata, 85100 Potenza, Italy; giovanni.salzano@unibas.it (G.S.); carmela.saturnino@unibas.it (C.S.)
*   Correspondence: s.sinicropi@unical.it; Tel.: +39-0984-493019
†   These authors equally contributed to this work.

**Abstract:** The carbazole class is made up of heterocyclically structured compounds first isolated from coal tar. Their structural motif is preponderant in different synthetic materials and naturally occurring alkaloids extracted from the taxonomically related higher plants of the genus *Murraya*, *Glycosmis*, and *Clausena* from the *Rutaceae* family. Concerning the biological activity of these compounds, many research groups have assessed their antiproliferative action of carbazoles on different types of tumoral cells, such as breast, cervical, ovarian, hepatic, oral cavity, and small-cell lung cancer, and underlined their potential effects against psoriasis. One of the principal mechanisms likely involved in these effects is the ability of carbazoles to target the JAK/STATs pathway, considered essential for cell differentiation, proliferation, development, apoptosis, and inflammation. In this review, we report the studies carried out, over the years, useful to synthesize compounds with carbazole moiety designed to target these kinds of kinases.

**Keywords:** carbazoles; heterocycles; STAT proteins; STAT inhibitors; target STATs; tumoral cells

## 1. Introduction

Over the years, sulphur and nitrogen-containing heterocyclic compounds have attracted particular interest. Many drugs, both of natural and synthetic origin, bear a heterocyclic structure (for example papaverine, theobromine, emetine, theophylline, atropine, codeine, reserpine, morphine, diazepam, chlorpromazine, barbiturates, and antipyrine) [1–6]. The pharmaceutical and pharmacological importance of these compounds resides in their ability to participate in hydrogen bonding with biological substrates (like specific proteins), where the heterocycle core can embody either H-acceptor as in heteroaromatic molecules or H-donor as in saturated *N*-heterocycles. This peculiarity is involved not just in pharmacological properties, but also in the pharmacokinetic behaviour of such drugs [7]. Among the various classes of heterocyclically structured compounds emerges the carbazole class (Figure 1), first isolated by Graebe and Glazer in 1872 from coal tar [8]. In 1965, Chakraborty et al. [9] reported the isolation and the activities of murrayanine (**2**) from Murraya koenigii Spreng. Since that moment, these compounds represented a great concern due to the attractive structural characteristics and encouraging biological effects displayed by several carbazole alkaloids [10–12].

Indeed, their structural motif is predominant in different artificial materials and naturally occurring alkaloids [12,13]. The majority of carbazole alkaloids were extracted from the taxonomically related higher plants of the genus *Murraya*, *Glycosmis*, and *Clausena* from the *Rutaceae* family. Moreover, these alkaloids were isolated from fungi and algae belonging to *Streptomyces*, *Aspergillus*, and *Actinomadura* species and the ascidian *Didemnum granulatum* [14]. A number of carbazoles derived from plants possess antitumor,

psychotropic, anti-inflammatory, antihistaminic, antibiotic, and antioxidant activities [15]. However, carbazole application is not limited to the biological and pharmaceutical field. In fact, due to their properties, they are also employed in the materials science field, as optoelectronic materials, conducting polymers, and synthetic dyes [16,17]. For instance, polyvinylcarbazoles (PVK) [18] have been thoroughly investigated for their employments in photorefractive materials and xerography. In fact, some poly (2,7-carbazole) derivatives (Figure 2) have been used in polymer solar cells [19]. The broad spectrum of their usage includes organic light-emitting diodes as green, red, and white emitters and some substituents at C-2, -3, -6, -7, and -9 positions can be responsible for these molecular and optical properties [20].

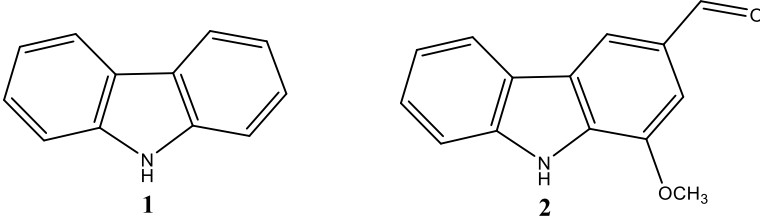

**Figure 1.** Structure of carbazole (**1**) and murrayanine (**2**).

**Figure 2.** Poly (2,7-carbazole).

Many natural carbazoles and their synthetic derivatives contain different substituents on the carbazole ring such as carbazomycin B (**4**) and carbazomadurin A (**5**), while others present a quadricyclic or pentacyclic structure, such as ellipticine (**6**) and staurosporine (**7**) respectively, or can appear as dimers, for example clausenamine A (**8**) (Figure 3) [21].

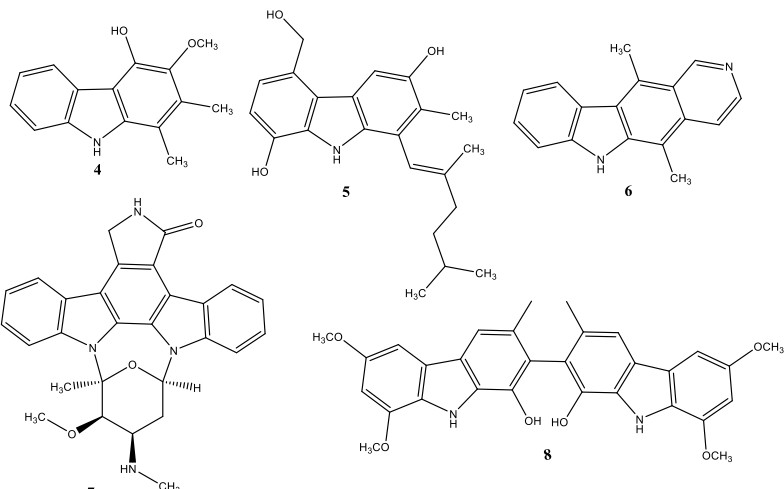

**Figure 3.** Structures of: carbazomycin B (**4**), carbazomadurin A (**5**), ellipticine (**6**), staurosporine (**7**), clausenamine A (**8**).

Due to the growing interest in carbazole bioactivities, different synthetic strategies have been set up and reported in the literature [14,22,23]. Knolker et al. [14] thoroughly reviewed the preparation methods of these carbazole alkaloids. Usually, the synthetic

routes to carbazoles include nitrene insertion, Fischer indolization, Pummerer cyclization, Diels-Alder reaction, dehydrogenative cyclization of diarylamines, etc. However, more recently, researchers explored the possibility of using transition metal-mediated C-C and C-N bond formation, cyclotrimerization, benzannulation, Suzuki–Miyaura coupling, and ring-closing metathesis [24–26].

Several studies aimed to compare the structural characteristics of natural carbazole alkaloids. They underlined that 3-methylcarbazole (**9**, Figure 4) could be the essential intermediate in their biosynthesis in higher plants. By contrast, 2-methylcarbazole (Figure 4) could be the communal biogenetic precursor to the tricyclic carbazoles isolated from lower plants. Nevertheless, the natural precursor of the carbazole nucleus has not yet been identified.

**Figure 4.** Structures of 3-methylcarbazole (**9**) and 2-methylcarbazole (**10**).

Concerning their biological activity, many research groups assessed the antiproliferative action of carbazoles against different types of tumor cells such as breast, cervical, ovarian, hepatic, oral cavity, and small-cell lung cancer [27–30]. One of the principal mechanisms possibly involved in this effect is the ability of carbazoles to target the signal transducers and activators of transcription (STATs) family of signaling pathways (especially STAT3). Indeed, this class of proteins is fundamental for the growth and survival of different human tumor cells [31,32]. In this context, the review aims to examine the studies carried out over the years to develop carbazolic compounds able to act on this target.

## 2. STAT Proteins: Novel Molecular Targets for Cancer Drug Discovery

Signal transducers and activators of transcription (STATs) constitute a group of cytoplasmic proteins that function as signal messengers and transcription factors engaged in critical cellular events with the use of multiple cytokines and growth factors. Since tyrosine has been phosphorylated, two STAT monomers develop dimers via mutual phosphotyrosine-SH2 interactions, migrate to the nucleus, and bind to STAT-specific DNA-response elements of target genes that cause gene transcription.

Concerning the biological functions of STATs, they contribute to cell differentiation, proliferation, development, apoptosis, and inflammation [33,34]. Indeed, STATs are typical transcription factors. They directly engage DNA regulatory elements (DREs) and thereby control the transcription of associated genes, linked to specific functions [35]. Up to now, seven STAT family members have been detected in mammals, designated as STAT1, STAT2, STAT3, STAT4, STAT5a, STAT5b, and STAT6.

The existing connection between the immune system cells and the JAK-STAT pathway has long been studied. Concerning the gain- or loss-of-functions mutation in genes encoding for JAK-STAs, immunodeficiency and susceptibility to infections are the most widespread phenotypes, demonstrating the crucial role of Jak–STAT signaling in the development and function of immune cells. STAT1 signaling, for example, suppresses type 17 immunity, which makes mice more susceptible both to bacterial and viral infections [36]. Indeed, it is involved in poxviruses, which encode interferon-receptor homologs; vaccinia viruses, which encode phosphatases; and Epstein–Barr viruses, encoding a protein that subverts multiple components of the interferon–STAT1 [35]. Additionally, STAT1 exerts a multidirectional antitumor effect. It blocks the cell cycle progression and angiogenesis and induces proapoptotic genes. Moreover, STAT1 signaling governs T helper type 1 (TH1) cell-specific cytokine production that changes both immune functions and inflammatory

responses by modifying the ratio between TH1 and TH2 cells. Furthermore, STAT1 has tumor-suppressing properties like TP53 and could be an antagonist for STAT3 and STAT5 activation. Besides, many studies corroborate the relationship between STAT1 downstream of interferons and NF-κB downstream of Toll-like receptors or the cytokine TNF, the two pathways can promote or 'prime' each other [35,37]. STAT2 signaling, instead, is important for its antiviral effects [38]. Further evidence indicates that altered STAT2 signaling may play a role in carcinogenesis by upregulating interleukin-6 (IL-6) production, which activates STAT3 [38].

Previous works demonstrated that in vitro cultured STAT3-deficient T cells did not react to IL-6 stimulation and could not be saved by IL-6 from apoptosis, demonstrating that STAT3's role is crucial for IL-6-mediated anti-apoptotic responses. Further, STAT3 altered functions in keratinocyte physiology could induce skin carcinogenesis. Moreover, constitutively active STAT3 has been identified in several malignancies, among which are breast, melanoma, prostate, head, and neck squamous cell carcinoma (HNSCC), multiple myeloma, pancreatic, ovarian, and brain tumors [39–43].

Aberrant STAT3 signaling stimulates tumorigenesis in part by modifying the expression of genes that rule the tumor growth processes. Examples are offered by genes encoding for p21WAF1/CIP2, cyclin D1, MYC, BCL-XL, BCL-2, vascular endothelial growth factor (VEGF), matrix metalloproteinase 1 (MMP1), MMP7 and MMP9, and survival [44].

A recent study demonstrated that unphosphorylated STAT3 possesses the function of promoting heterochromatin formation in lung cancer cells, suppressing cell proliferation in vitro, and suppressing tumor growth in mouse xenografts [45]. Furthermore, transforming growth factor-α (TGF-α)-mediated epidermal growth factor receptor (EGFR) signaling plays a vital role in the activation of STAT3 in some head and neck cancer cell lines [46]. In addition, STAT3 mutations were identified in multiple peripheral T-cell lymphomas (PTCL), along with high pY-STAT3 expression, particularly in angioimmunoblastic T-cell lymphoma (AITL) and anaplastic large cell lymphoma (ALCL) patient samples, two different PTCL cancer types [47]. By modulating the genetic and pharmacological profile of persistently active STAT3, it is possible to hamper the tumor progression *in vivo*. STAT4 is also essential for IL-12 function, which governs the differentiation of TH1 cells and their inflammatory responses [48]. Therefore, STAT4 signaling is related to autoimmune diseases [49].

STAT5 is present in two isoforms, STAT5a and STAT5b [50]. Normal STAT5 signaling is crucial in mammary gland development, milk production, and hematopoiesis [51]. Constitutive STAT5 triggering is also involved in the development of HNSCC, chronic myelogenous leukemia (CML) [52], and breast, prostate, and uterine cancers [53]. In particular, the effects could be related to the aberrant STAT5 activation by BCR–ABL in CML43 protein. Indeed, it is extensively accepted that STAT5 and STAT3 have in common equal functions in promoting cancer, inducing the expression of pro-proliferative and anti-apoptotic genes [54,55]. Mice lacking both STAT5A and STAT5B showed a large variety of immunological defects and died soon after birth. In addition, the lineage-restricted deletion of STAT5 has been key in defining its in vivo functions, including the role in CD4$^+$ T cell differentiation, NK-cell-mediated immunosurveillance, CD8$^+$ T cell memory187, DC-lineage specification and DC-driven type 2 inflammation. [35] In addition, STAT5A and B are implicated in chronic myeloid leukemia. In fact, the transfection of K-562 leukemia cells either with anti-STAT5A or anti-STAT5B, led to a greatly increased apoptosis rate [56].

IL-4 and IL-13 trigger STAT6 signaling that sustains immune function and regulates the equilibrium between inflammatory and allergic immune responses [57,58]. STAT6 signaling also supports luminal mammary epithelium progression and is involved in the pathology of lung and airway diseases, [59–61]. A recent study assessed that E2F1 is essential for maintaining the level of signal transducer and activator of STAT6 in HCT116 colorectal cancer cells. Mechanistically, E2F1 induced specificity protein 3 (SP3) directly binds to the promoter of the STAT6 gene and activates its transcription in CRC cells. As a

result, it was demonstrated that the E2F1/SP3/STAT6 axis is required for the IL-4-induced epithelial-mesenchymal transition of colorectal cancer cells [62].

Linked to STAT proteins are Janus kinases (JAK1, JAK2, JAK3, and TYK2) enzymes, structurally related to each other, with tyrosine kinase activity assisting in the passage of signals from the cell surface to the inside. The JAKs pathways are implicated in the onset of many inflammatory and autoimmune pathologies, among which are rheumatoid arthritis (RA), psoriasis, and inflammatory bowel disease. The main reason is due to the use of the JAK-STAT3 pathway by a substantial amount of cytokines and hormones for intracellular signaling. Indeed, in psoriasis, diverse cytokines implicate the JAK-STAT pathway, such as type I and II IFN, IL-12, IL-22, IL-23, and IL-13. In particular, JAK activation starts, at first, upon ligand-mediated receptor multimerization because two JAKs are brought into proximity, allowing trans-phosphorylation. The subsequent step includes the phosphorylation of a conserved tyrosine residue near the C-terminus of STATs by activated JAKs [63]. Concerning the JAKs family, JAK1, JAK2, and TYK2 expression can be detected throughout the whole body. On the contrary, JAK3 has been specifically identified in the hematopoietic cells. JAK1 displays its function by conveying the signals of numerous proinflammatory cytokines and cooperates with JAK3 in lymphopoiesis by binding to heterodimeric interleukin (IL) receptors. JAK2 is one of the major mediators in the signal transmission of hematopoietic factors. Moreover, recently, a gain-of-function point mutation in the pseudokinase domain (JH2) of JAK2 has been ascertained in myeloproliferative neoplasm (MPN) patients [64]. JAK3 and TYK2, instead, possess immunomodulatory functions [65].

## 3. STATs Inhibitors

### 3.1. Peptides and Small Molecules as STATs Inhibitors

Activation of STAT1, STAT3, and STAT5 is highly frequent in almost all tumors studied, with a higher incidence of abnormal STAT3 activation [66]. Starting from the breakthrough of the first peptide inhibitor of a STAT protein, diverse strategies to counteract against STAT signaling have been pursued. An example is provided by small-molecule dimerization disruptors (SMDDs) or phospho-peptidomimetic inhibitors (PPMIs) targeting the phospho-Tyr-SH2 domain interaction at the interface of dimers of STAT proteins. The main consequence consists of the disruption of STAT–STAT dimers. STAT–SMDD or STAT–PPMI forms heterocomplexes, suppressing STAT signaling and function. On the other hand, the bond between growth factors and cytokines and their receptors on the cell surface activates STAT tyrosine phosphorylation. Tyrosine kinases responsible for STAT phosphorylation are the targets of small-molecule tyrosine kinase inhibitors. These modulators can prevent the induction of STAT phosphorylation and signaling. They act by impeding STAT dimerization using peptides or peptidomimetics identified through different processes as structure-based design, small molecules identified by molecular modeling, virtual or library screening, or natural products [66].

Other methods, comprise oligodeoxynucleotide (ODN) decoys (peculiar STAT DNA-binding domain inhibitors) and antisense oligonucleotides (ASOs) that interfere with STAT mRNA. Studies on the molecular basis of oncogenesis, regarding oncoproteins like v-Src, report alterations in intracellular signaling proteins involved in several malignancies. The discovery that STAT3 is constitutively activated in v-Src transformation indicated the possible central role of STATs in oncogenesis [67,68]. Additionally, other transforming tyrosine kinases such as v-Eyk [69], v-Ros [70], v-Fps [71], Etk/ BMX [72], and Lck [73], activate STAT3 in the oncogenic process. Constitutive STAT3 activation is also connected with the transformation caused by tumor viruses, among which are HTLV-1 [74], polyomavirus middle T antigen [75], EBV [76], and herpesvirus saimiri [73], that directly or indirectly activate JAKs or Src family tyrosine kinases.

For the reasons explained above, different therapeutic agents, targeting aberrantly active STAT3, could affects several human cancers making STAT3 the target in many drug discovery research efforts. STAT3 drug discovery research concentrated on targeting the pTyr-SH2 domain interaction [77,78] due to its relevance in promoting STAT3 dimeriza-

tion and function. A semirational, structure-based design study determined the first SH2 domain-binding peptides and peptidomimetics that break up the STAT3 pTyr-SH2 domain interactions and STAT3–STAT3 dimerization [78,79]. The native parent pTyr peptide, PY*LKTK (where Y* stands for pTyr) and its modified forms prevented the DNA-binding and transcriptional activities of STAT3 at high doses [79]. Peptidomimetic and non-peptide analogues, such as ISS-610 and S3I-M2001, counteracted STAT3 activity in vitro and accommodated aberrantly active STAT3 [78,79]. In particular, S31-M2001 constrained the progression of human breast tumor xenografts, while ISS-610 blocked cell growth and triggered apoptosis in vitro [66].

Moreover, phosphopeptide binding sequences with the primary structure pTyr-Xxx-Xxx-Gln (where Xxx represents any amino acid) impeded STAT3 activation. They derived from leukemia inhibitory factor (LIF), IL-10 receptor, epidermal growth factor receptor (EGFR), granulocyte colony-stimulating factor (GCSF) receptor, or glycoprotein 130 (gp130) [80,81]. In this research, the peptidomimetic Ac-pTyr-Leu-Pro-Gln-Thr-Val-$NH_2$ inhibited STAT3 activity ($IC_{50}$ values of 150 n$\mu$M) [81,82]. Furthermore, a 28-mer native peptide identified as SPI, obtained from the STAT3 SH2 domain, hampered the STAT3 pTyr-SH2 domain interaction and signaling. It suppressed cell viability and caused programmed cell death of human breast, pancreatic, prostate, and non-small cell lung cancer (NSCLC) cells in vitro [83].

In silico studies of chemical libraries detected several small-molecule inhibitors of STAT3 activity. The main mechanism regards the disruption of STAT3–STAT3 dimerization. STA-21 (or NSC628869) was recognized from the screening of the National Cancer Institute (NCI) chemical library as an inhibitor of STAT3 dimerization, DNA-binding activity, and transcriptional function in breast cancer cells at 20 $\mu$M [84]. Its structural analogue, LLL-3 (with enhanced membrane permeability), reduced cell viability in vitro and intracranial tumor size in vivo in glioblastoma animal models [85]. Moreover, a catechol (1,2-dihydroxybenzene) compound was detected from Wyeth's proprietary small-molecule collection as a STAT3 SH2 domain inhibitor, active at 106 $\mu$M against a multiple myeloma cell line [86]. In addition, S3I-201 (or NSC74859) as a STAT3–STAT3 dimerization disruptor active at 60–110 $\mu$M [87] also emerged from the NCI chemical library. S3I-201 hampered STAT3 DNA-binding and transcriptional activities, inhibited cellular growths, and promoted the apoptotic process of tumoral cells harboring constitutively active STAT3, and suppressed the growth of human breast cancer xenografts [87]. Several of its derivatives, such as S3I-201.1066, BP-1-102, and S3I-1757, exhibited a better efficacy, with $IC_{50}$ values of 35 $\mu$M (S3I-201.1066), 6.8 $\mu$M (BP-1-102), and 13.5 $\mu$M (S3I-1757), and suppressed cell growth, malignant transformation, survival, migration and invasiveness in vitro of malignant cells harboring aberrantly active STAT3.

Moreover, Cpd30 (4-(5-((3-ethyl-4-oxo-2-thioxo-1,3-thiazolidin-5-ylidene)methyl)-2-furyl)benzoic acid) (Figure 5) moderately blocked STAT3, prevented STAT3 nuclear translocation upon IL-6 stimulation, and triggered apoptosis in breast cancer cells harboring constitutively active STAT3 [54]. Its analogue, Cpd188 (4-((3-((carboxymethyl)thio)-4-hydroxy-1-naphthyl)amino)sulphonyl)benzoic acid) (Figure 5), combined with docetaxel, lowered tumor growth in chemotherapy-resistant human breast cancer xenograft models.

From another virtual screening of small molecules emerged Stattic, a non-peptide small molecule able to target the STAT3 SH2 domain and inhibit STAT3 signaling at 10 $\mu$M [88]. It triggered apoptosis of STAT3-dependent breast cancer and neck squamous cell carcinoma HNSCC cells and arrested growth of orthotopic HNSCC tumor xenografts [88]. STX-0119, STAT3 SH2 domain antagonist generated antitumor effects both in vitro and in vivo in a human lymphoma model, probably by disrupting STAT3–STAT3 dimerization, with a modest activity on STAT3 phosphorylation [89]. Fragments of STX-0119 and stattic were chemically combined to create HJC0123 (Figure 5) [90], which impeded STAT3 phosphorylation and transcriptional activity in breast cancer cells and possessed antiproliferative effects towards breast and pancreatic cancer cells in vitro ($IC_{50}$ values of 0.1–1.25 $\mu$M).

**Figure 5.** Some inhibitors of STAT3 activity with no carbazole structure.

The oral administration of HJC0123 blocked the growth of human breast cancer xenografts [90]. Furthermore, OBP-31121 inhibited STAT1, STAT3, and STAT5 phosphorylation. OBP-31121 reduced cell proliferation and triggered apoptosis in vitro whereas it prevented tumor growth in vivo in gastric cancer models, and it further sensitized gastric cancer cells to cisplatin and 5-fluorouracil [91].

*3.2. Natural Compounds as STATs Inhibitors*

Several studies also identified many lead candidates such as STAT3 inhibitors from natural products. However, the mechanism of action is not completely defined. Curcumin, a phenolic compound derived from the perennial herb *Curcuma longa,* hampers JAK–STAT signaling at 15 μM, promotes cell cycle arrest, and impedes cell invasion in vitro in a small cell lung cancer model [92]. The treatment of mice bearing gastric cancer xenografts with curcumin suppressed IL-6 production by IL-1β-stimulated myeloid-derived suppressor cells, associated with decreased activation of STAT3 and nuclear factor-κB (NF-κB).

Curcumin analogues with ameliorated bioavailability and stability, such as FLLL32, had improved efficacy (IC$_{50}$ values of 0.75–1.45 μM) in inhibiting both pSTAT3 and total STAT3. Furthermore, they caused STAT3 ubiquitylation and possible proteasomal degradation in canine and human osteosarcoma cells in vitro [93]. HO-3867 (Figure 5), curcumin analog, downregulated likewise STAT3 signaling in cisplatin-resistant human ovarian cancer cells, thus intensifying sensitivity to cisplatin [94]. HO-3867 also promoted apoptosis in BRCA1-mutated human ovarian cancer cells harboring aberrantly active STAT3 [95].

Other findings estimated that LLL12 (Figure 5), another small-molecule inhibitor of STAT3 signaling based on curcumin [96], hampers STAT3 activation by impeding its recruitment to the receptor and avoiding phosphorylation by tyrosine kinases, and by preventing the dimerization [97]. LLL12 decreased cell viability, promoted the apoptotic process, and suppressed colony formation and migration in vitro in glioblastoma, osteosarcoma, and breast cancer cells. It also interfered with angiogenesis, tumor vasculature development, and tumor growth in vivo in osteosarcoma xenograft models [96,97]. Resveratrol (3,5,4′-trihydroxystilbene) (Figure 5), a polyphenolic compound mainly found in red grapes, inhibits STAT3 signaling at high micromolar concentrations, thus affecting tumoral cell growth. Resveratrol treatment also prevents constitutive and IL-6-induced STAT3 activation in multiple myeloma, leukemia, and other tumor cell types, reduces the expression of BCL-2 and other anti-apoptotic proteins, in vitro [98]. The resveratrol derivative LYR71 arrested cell growth (IC$_{50}$ value of 20 μM) and decreased STAT3-mediated MMP9 expression [99]. Moreover, the inhibition of the JAK–STAT3 pathway by resveratrol or its analogue, piceatannol (3,3′,4,4′-transtrihydroxystilbene), reduced BCL-XL and BCL-2 expression and sensitized lung carcinoma, multiple myeloma, prostate pancreatic

cancer, and the glioblastoma multiforme patient-derived CD133-positive cells to radiation or chemotherapy in vitro.

### 3.3. Carbazoles as STAT Inhibitors

Psoriasis, a common inflammatory skin disease, is treated, according to its severity, with a large panel of therapies associated with side effects. Coal tar represented an ancient treatment of psoriasis. To assess the potential mechanism of action, Arbiser et al. in 2006 [31] fractionated coal tar through chromatography determining carbazole as the active compound. STAT3 inhibition resulted in one of the main mechanisms implied in its antiangiogenic activity. Indeed, STAT3 is involved as a signaling pathway in psoriasis. Moreover, carbamazepine (Figure 6) (an anticonvulsant structurally related to carbazole) exhibited great potentiality in patients with generalized psoriasis. For this reason, this research group focused their attention on carbazole **1** included in coal tar.

**Carbamazepine**

**Figure 6.** Structure of carbamazepine, an anticonvulsant structurally related to carbazole.

Furthermore, carbazole inhibited IL-15, whose level is increased in psoriasis, and decreased the effects of inducible nitric oxide synthase (iNOS). STAT3, IL-15, and iNOS need rac GTPase activation to exert their activities. Thus, the ability of carbazole to inhibit rac activation has been demonstrated, stating that this could be the origin of its ability to reduce inflammation and counteract angiogenesis [31]. Starting from the assumption that STAT3 is one of the main mediators implied in the immune and angiogenic features of psoriasis, Arbiser et al. [31] went into the possible actions of carbazole and 2-hydroxycarbazole (a carbazole metabolic by-product) toward the STAT3-mediated transcription. Their data, obtained on RAW264.7 macrophages, indicated that both carbazole and 2-hydroxycarbazole (**11**, Figure 7) could downregulate STAT3-mediated transcription.

**11**

**Figure 7.** Structure of 2-hydroxycarbazole (**11**).

They assessed the specificity against STAT3, since carbazole possessed no activity on a constitutive promoter. Activity on STAT3 phosphorylation was excluded as a potential carbazole mechanism of action. Indeed, no effect on STAT3 phosphorylation was detected, implicating the existence of an alternative mechanism. A fundamental prerequisite to activate STAT3 is rac activation. In fact, small GTPase rac influences STAT3 transcription, IL-15 production, and iNOS activity. Obtained results outlined that carbazole markedly hampered rac activation by VEGF in HUVEC cells.

In 2013, Sarkar et al. [100] analyzed the molecular mechanism held responsible for the antiproliferative activity of mahanine (**12**, Figure 8) toward several pancreatic cell lines. In fact, natural compounds have been demonstrated to hold a great deal of promise as antineoplastic agents. In particular, the purified carbazole alkaloid mahanine, isolated

from the edible plants *Murraya koenigii* and *Micromelum minutum*, displayed antimutagenic, antimicrobial, and cytotoxic activities [101].

**Figure 8.** Structure of mahanine (**12**).

Furthermore, mahanine triggered the apoptotic process in histiocytic lymphoma, promylocytic leukemia, and prostate cancer cells [102–104]. At first, it was confirmed that mahanine induced apoptosis in an in vitro model of the human pancreatic ductal adenocarcinoma cell line, MIAPaCa-2 ($IC_{50}$ amounting to 13.9 μM). Subsequently, the possible effects of mahanine on the molecular chaperone Hsp90 have been studied, starting from the assumption that it presents high expression levels in several tumor cells and regulates different client proteins, including STAT3. The results underlined that mahanine triggers the accumulation of ROS in both a time- and concentration-dependent manner in MIAPaCa-2 cells. The main consequence of this activity regards the oxidative insult of Hsp90, mainly involved in maintaining folded proteins in their proper conformation. Immunostaining studies demonstrated the depletion of the two Hsp90 client proteins Akt and STAT3 in orthograft pancreatic adenocarcinoma mouse model treated with mahanine. Furthermore, mahanine lowered, in a concentration-dependent way, other client proteins such as B-Raf, A-Raf, mutated p53, GSK3b, and PKCb at 24 h in MIAPaCa-2 and AsPC-1 up to a dose of 30 μM. Besides, the authors proved to decrease mahanine concentration (10–20 μM), demonstrating its efficacy in MIAPaCa-2 cells after 24 h of treatment. In fact, mahanine diminished the protein level of Akt and CDK-4 at 15 μM, while B-Raf, STAT3 and Bcl-XL were conspicuously reduced at 17.5 μM. As previously mentioned, the molecular chaperone Hsp90 has a crucial function in the survival of cancerous cells in which it is extensively expressed. As a result, this makes it a promising target for chemotherapeutic agents. Indeed, it mediates the folding, assembly, and maturation of many client proteins, among which are HER2, EGFR, PI3K, Akt, B-Raf, STAT3, GSK3b, Cdk4, mutated p53, and steroid receptors that participate in malignant cancer development [105].

In silico studies demonstrated the binding capability of mahanine to Hsp90 via noncovalent weak interactions, in a polar pocket, apart from the ATP-binding cavity. Mahanine constituted two hydrogen bonds with Hsp90, the first with the side chain oxygen of Glu47 through its NH group and the second with the side chain oxygen of Asn51 through its OH group. Moreover, van der Waals interactions with Arg46, Ile43, Gly132, Gln133, Met130, Ser129, Phe138, Ile131, and Gly137 were established. The most advantageous configuration of the ligand with the protein had a binding energy of −7.6 kcal mol$^{-1}$, with a micromolar binding affinity ($K_D$ = 3.16 μM) [100].

In 2013, Saturnino et al. [32], in their preliminary study, synthesized a series of *N*-alkylcarbazole derivatives to evaluate their potential STAT3 inhibitory activity. Their studies, indeed, highlighted that a crucial role in modulating the lipophilic properties of the carbazole was played by the substituent in N9. They analyzed a small series of carbazoles *N*-alkylated with C5, C6, and C7 alkyl chains [106]. Additionally, the alkyl chains were functionalized with dimethyl 5-hydroxyisophthalate (for derivatives **13**) or methyl salicylate (for derivatives **14**) as substituents (Figure 9). Their findings highlighted the ability of compounds **13a-c**, among others, to inhibit STAT3 phosphorilation and its nuclear translocation in acute monocytic leukemia at 50 μM with different potencies, amounting to 50%, 90%, and 95%, respectively. A crucial element for this effect seems to be the length of an alkyl linker inserted in these compounds. They obtained these outcomes by performing

EMSA and Western blot analysis in THP-1 cells treated with IL-6 (20 μg/mL) for 15 min. Indeed, IL-6 augmented the STAT3 DNA binding activity as indicated by EMSA/supershift experiments with anti-STAT3 antibody. Furthermore, obtained data suggested a time- and dose-dependent effectiveness for **13a-c**. These compounds, however, could not block IFNy-induced STAT1 nor TNF-α LPS-induced NF-kB activation. Phosphorylation on specific tyrosine residues and consecutive translocation into the nucleus are crucial to activate STATs. Their studies assessed that compounds **13a-c** lowered, at different levels, IL-6-induced tyrosine705 phosphorylation of cytosolic STAT3 without impacting the whole STAT3 protein. Concerning the structure-activity relationship, the insertion of an *N*-alkyl chain added an interaction point to the target protein.

**13a** n= 5
**13b** n= 6
**13c** n= 7

**14a** n= 5
**14b** n= 6
**14c** n= 7

**Figure 9.** Structure of: hydroxyisophthalate derivatives (**13**) and methyl salicylate derivatives (**14**).

Slight changes of its terminal phenyl ring significantly modified the activity profile (compare **13a-c** with **14a-c**). Another important feature to improve the activity is the length of the alkyl linker as observed by comparing **13a** with **13b** and **13c**) [15,32].

Although prior research substantiated that [31,32] some carbazoles affected the STAT3-mediated transcription and/or STAT3-DNA binding and phospho-STAT3 at high doses (30–50 μM), little evidence regards the molecular mechanisms involved in this process. Hou et al. [107] have been previously fascinated by the possibility of targeting STAT3 pathways. They, indeed, tried to identify therapeutic agents against cancer. They focused on the development of fluorescent small molecules. The silver lining of these compounds was the capability to be observed directly as it interacts with receptor-positive cell lines [108–110]. However, a preceding survey assessed that the antiproliferative activity of the free carbazole on the cancer cell lines is lower than the corresponding scaffolds with the dansyl group. In view of the above, this research group synthesized a small series of carbazole derived compounds with fluorophore (Figure 10). Among them, compound **15** was the most effective compound.

**Figure 10.** General structure of carbazoles fluorophore and structure of 7-hydroxy-1-methyl-9H-carbazol-2-yl 5-(dimethylamino)-naphthalene-1-sulfonate (**15**).

It inhibited the STAT3-mediated transcription and IL-6-induced phosphorylation of STAT3 in triple negative breast cancer (TNBC) cells. Indeed, they examined the antiproliferative effects of **15** on a panel of breast cancer cells, finding the best results for the invasive ductal carcinoma cells SUM1315MO2. Moreover, **15** suppressed cell proliferation in A431 (a squamous carcinoma cell line), A549 (a lung cancer cell line) and PC-3 (a prostate cancer cell line) with GI50 values 0.16 μM (for A431), 2.5 μM (for A549), and 3 and 7.9 μM (for PC-3), respectively.

According to the inhibition of STAT-3, the cell treatment with compound **15** provoked a reduction of cyclin D1 levels, transcriptional target of STAT3. Besides, in vitro and in vivo studies demonstrated that the inhibitory effects of **15** on phospho-STAT3 were via the up-regulation of cytoplasmic protein−tyrosine phosphatase PTPN6. This was measured on the two lines of invasive ductal carcinoma HS578T, and SUM149PT. Further analysis assessed a cytoplasmic fluorescence of compound **12** in MDA-MB-231 cells, evidencing that the compound enters human breast cancer cells. Compound **15**, in fact, triggered apoptosis in breast cancer cell lines in vitro and it was efficient at inhibiting the in vivo growth of human TNBC xenograft tumors (SUM149) with a 57% reduction of tumor volume on the fifth day without showing any toxicity. Furthermore, compound **12** also suppressed the growth of human lung tumor xenografts (A549) harboring aberrantly active STAT3. The maximum inhibitory effect was achieved on the ninth day (inhibition rate 73.76%) [107].

Subsequently, in 2017, the same research group focused their attention on this compound deepening the knowledge of the molecular mechanism [111]. Indeed, as just reported, in their preceding work they discovered that **15** was able to inhibit phospho-STAT3 (Y705) by induction of PTPN6/SHP-1 expression. However, previous studies underlined that PTPN6/SHP-1 is epigenetically repressed by STAT3- DNMT1 [112] and acetylation of STAT3 (K685) is critical to bind DNA (cytosine-5)-methyltransferase 1 (DNMT1). They found that **15** not only decreased the phospho-STAT3 (Y705) but also the acetyl-STAT3 (K685) levels in HS578T cells, both in a dose-dependent manner.

By using Western blot experiments, they highlighted that STAT3 binds to DNMT1 in HS578T cells. However, **15** inhibited the capability of forming this interaction at 0.3 μM. The disruption of STAT3-DNMT1 interaction by **15**, instead, happened at 0.1 μM. This effect was achieved also in SUM1315MO2 cells. In contrast, the reference molecules used, the JAK2 inhibitor (AZD1480) or JAK1/2 inhibitor (CP690550) did not impact STAT3-DNMT1 interaction. Moreover, they established that the disruption of STAT3- DNMT1 interaction by **15** is dependent on the deacetylation of STAT3 at K685. Furthermore, starting from the assumption that the STAT3-DNMT1 interaction is important for DNA methylation in the promoter region of TS genes, they identified four TS genes that were demethylated by **15** in HS578T cells, among which the retinoic acid receptor beta (RARB), neurogenin 1 (NEUROG1), PDZ and LIM domain 4 (PDLIM4), and Von Hippel–Lindau tumor suppressor (VHL). Then, they evaluated the methylation status of these promoters in two TNBC cells. It has been discovered that the promoter regions of VHL and PDLIM4 genes were highly methylated and **15** near commonly demethylated these promoters. Demethylation of VHL and PDLIM4 gene promoters by **15** resulted in the reactivation of mRNA expression of these genes in both HS578T and MDA-MB-231 in a dose-dependent manner. At the end, they carried out in vivo studies, highlighting that **15** significantly reduced tumor growth without inducing loss of body weight [111].

In 2015, Cuenca-López et al. [113] studied the antitumor properties of **16** (Figure 11), a hybrid indolocarbazole analog produced by combinatorial biosynthesis of Rebeccamycin and Staurosporine genes.

The authors outlined the antiproliferative activity of this compound towards HS578T, BT549, MDA-MB-231, and HCC3153 with IC$_{50}$ values in the nanomolar range. Moreover, **16** was able to reduce colony formation of these cells and decrease tumor volume in mice. Besides, their main purpose was to investigate the molecular mechanism of **16** by analyzing its effects on kinases profile in TNBC. Through biochemical experiments, they proved that **16** suppressed downstream components of the PI3K/AKT pathway including

AKT (phosphorylated at T308 and S473) and pS6. In the same way, it suppressed p-Stat3 and p-Stat1 in HS578T and BT549 cells. To deepen their knowledge of the antiproliferative mechanism, they carried out some experiments on cell cycle and apoptosis. Their findings suggested that it caused an accumulation of cells HS578T, BT549, and MDA-MB-231 in G2/M phase at 24h whereas further analysis underlined that the apoptotic mechanism was caspases independent. Therefore, to examine if **16** induced DNA damage, they evaluated the phosphorylated γH2AX levels, a protein necessary for checkpoint-mediated cell cycle arrest and DNA repair following double-stranded DNA breaks. Treatment with **16** in HS578T, BT549, and MDA-MB-231 exhibited an enhancement in the phosphorylated levels of γH2AX at early time points. They also noted that **16** provoked the phosphorylation of p53 and Chk2, validating the induction of DNA damage. In addition, they tried to explore the effects of **16** when associated with chemotherapeutic agents used in the clinical setting for triple negative tumors such as vinorelbine, docetaxel, and carboplatin. Their results displayed a synergistic interaction of **16** with all these compounds, mostly with docetaxel, in HS578T, BT549, and MDA-MB-231 cells and improved effects *in vivo*. In addition, **16** possessed a good pharmacodynamic and pharmacokinetic profile. Indeed, it exhibited a clear decrease of pS6 and pSTAT1 at 30 min in vivo in the extracted tumors. pSTAT3 was also moderately impaired. Induction of pγH2AX was reached at 60 min. Moreover, they evaluated the concentration of the drug in two resected tumors per time point, assessing a time-dependent accumulation, reaching over 1000 ng/g. The obtained results underlined that targeting acetylation of STAT3 (K685) by small molecule inhibitors could be a valid therapeutic option for cancer treatment. Acetylation of STAT3 and DNMT1 present high levels in a wide range of malignancy. Conceivably, in these diseases, the STAT3-DNMT1 complex is engaged in the repression of TS genes by DNA methylation. With this in mind, an in-depth investigation of acetyl-STAT3 and DNMT1 status in human cancers could offer a potential therapeutic opportunity with the aim of targeting the STAT3-DNMT1 interaction [113].

**Figure 11.** Structure of (5*R*,7*S*,9*S*)-7,8-dihydroxy-9-methyl-6,7,8,9-tetrahydro-5*H*,14*H*-17-oxa-4b,9a,15-triaza-5,9-methanodibenzo[b,h]cyclonona[jkl]cyclopenta[e]-as-indacene-14,16(15*H*)-dione (EC-70124, **16**).

Oscar Estupiñan et al. in 2019 [114] evaluated the anticancer effect of **16**, too. They assessed that PI3K/AKT/mTOR, JAK/STAT or SRC were the main activated pathways in cell-of-origin sarcoma models and/or sarcoma primary cell lines. Compound **16** blocked the phosphorylation of these targets and impeded proliferation triggering the DNA damage, cell cycle arrest, and apoptosis. Moreover, **16** in part decreased tumor growth *in vivo*. Furthermore, it decreased the expression and the effect of ABC efflux pumps implicated in drug resistance. Data also suggested that combining compound **16** with doxorubicin resulted in a synergistic cytotoxic effect in vitro and an enhanced antitumor effect in the animal model. They found that **16** is much more successful than midostaurin in preventing the phosphorylation of ERK1/2 at Thr202 and Tyr204, AKT at Ser473 and Thr308, pS6 at Ser235 and Ser236 and 4EBP at Ser65, which are the kinases/kinase substrates most enabled

among a panel of relevant signaling molecules in a cell model of MRCLS. Furthermore, **16** prevented the activation of SRC, STAT1, and STAT3, activated in several primary sarcoma cell lines. The multikinase inhibitory effect of **16** in sarcomas is linked to a potent antiproliferative activity if compared with midostaurin. This is due to the induction of DNA damage followed by a more stringent S-phase arrest and apoptosis. In addition, it is known that midostaurin and other indolocarbazoles inhibit Aurora kinase abrogating the mitotic spindle checkpoint and the accumulation of cells with 4 N and 8 N DNA content [114].

In 2015 Botta et al. [115] synthesized a series of 1,4-dimethyl-carbazole derivatives (Figure 12) and tested them for their potential activity on STAT3.

**Figure 12.** 1,4-Dimethyl-carbazole derivatives (**17–26**).

The design of molecules has been assisted by in silico studies with the purpose to identify new ligands of SH2 domain of STAT3 responsible for the protein dimerization, where a monomer identifies the Pro-Tyr/pTyr-Leu-Lys-Thr-Lys sequence of its partner. Indeed, the in silico analysis was performed by selecting the protein region accommodating the Pro-Tyr/pTyr-Leu-Lys-Thr-Lys sequence and especially the interactions involving the Tyr705/pTyr705, essential for the possible ligand binding. Therefore, the possible effects on the cellular viability of human melanoma (A375) and human epithelial cervix adenocarcinoma (HeLa) cell lines, which constitutively express STAT3, have been evaluated. Their findings evidenced that 1,4-dimethyl-carbazole (**17**) (IC$_{50}$ of 80.0 µM) had similar activity to doxorubicin (IC$_{50}$ of 87.0 µM), used as a reference molecule, toward A375 cells.

Compounds **18**, **20**, **24**, and **26** achieved a similar trend of results on both cell lines in comparison with doxorubicin whereas compounds **19** and **21** exhibited IC$_{50}$ values on A375 of 50 and 60 µM, respectively. Moreover, the most active compounds **19–21** displayed a considerable inhibition rate at 72h of STAT-3 expression with respect to the control cells, with a percentage of inhibition amounting to 94.33, 90.66, and 91.0 %, respectively. In silico studies revealed that the position 6 of carbazole ring faces the cavity accommodating the Tyr705/pTyr705. Specifically, this research group verified that the Tyr705/pY705 is in a small hollow, to interact with the side chains of Lys591, Arg609, Ser611, Glu612 and Ser613. For this reason, they designed compounds **18–22** inserting at C-6 a hydrogen bond acceptor to replicate this interaction. Furthermore, they added hydroxyl, methoxy, ethyl ester groups, and a chlorine and a sulfonamide function. Since position 3 of carbazole is close to the side chain of Arg595, they explored the C-3 position to ameliorate the binding to the protein. Therefore, in light of the simple introduction of a nitro group able to interact

with Arg595, compounds **23** and **24** have been designed. Furthermore, compounds **25** and **26** attempted to raise the contacts with STAT3 through van der Waals interactions by introducing alkyl groups on the nitrogen to the detriment of the hydrogen bond. At the end, the designed compounds **18–26** were docked into the binding cavity of STAT3. The docked poses of **18–26** accommodated well into the protein binding cavity and seem to respect the predicted interactions [115].

Since JAK and STAT proteins (as previously stated) are connected to a pathway that involves many cytokines and hormones for intracellular signaling with potential involvement in cancer, Zimmermann et al. [116] designed and synthesized a series of 9*H*-carbazole-1-carboxamides (Figure 13) in order to evaluate their potentiality as selective ATP-competitive inhibitors of Janus kinase 2 (JAK2).

**Figure 13.** Structure of 9*H*-carbazole-1-carboxamides (**27–29**).

By means of in silico studies, they tried to optimize their lead compound **27**, showing a good JAK2 inhibitory potency but poor selectivity toward JAK family. The (*S*)-dimethylamino-pyrrolidine amide **28** exhibited an enhanced selectivity against other members of the JAK kinase family. Moreover, **28** was tested in vitro for its antiproliferative effect on SET-2 cells (adult acute megakaryoblastic leukemia) reaching an $IC_{50}$ value of 80 nM. Further studies were aimed to assess its in vitro ADME properties. Data suggested that **28** possessed a greater solubility than compound **27** (>1.9 mg/mL at pH 1.0; 0.41 mg/mL at pH 6.5; compared to <0.001 mg/mL across pH range for 1). Another compound with a significant selectivity on JAKs was **29**, particularly active against JAK2 ($IC_{50}$ amounting to 5.5 nM), which also displayed an antiproliferative activity against SET-2 cells with an $IC_{50}$ value amounting to 130 nM. Concerning the metabolic stability of **29**, it is similar to **28**, except its solubility at acidic pH was enhanced when compared to the initial screening (0.044 mg/mL at pH 1.0) [108].

Based on the knowledge that phosphorylation of STAT3 at Y705 induces the dimerization and translocation into the nucleus to relay the oncogenic signals by expressing the genes implicated in proliferation, antiapoptosis, angiogenesis and tumor evasion in 2016, Baburajeev et al. [117] reported the synthesis and evaluation of substituted carbazole derivatives (Figure 14) using nano-cuprous oxide as a catalyst via intramolecular C–N bond forming reactions. Among them, (3-acetyl-6-chloro-9*H*-carbazol-9-yl)methyl)-[1,10-biphenyl]-2-carbonitrile (**30**) exhibited the greater antiproliferative activity towards two lung cancer cell lines: A549 and LLC, establishing itself as a lead compound. The calculated $IC_{50}$ values amounted to 13.6 and 16.4 μM, respectively for the two cell lines.

**Figure 14.** Carbazol-carbonitrile derivatives (**30–32**).

Paclitaxel, used as a reference molecule, instead achieved IC$_{50}$ of 0.0044 μM against A549 cells. Furthermore, they performed their studies on A549, HCC-2279 and H1975 cells, treating them with 10 μM of ACB for 6 h to examine the effect on the persistent activation of STAT3 by Western blot analysis using antibodies that recognize phosphorylation of STAT3 at Tyr-705. They found that **30** inhibited STAT-3 phosphorylation in A549, HCC-2279 and H1975 cells at a concentration of 10 μM. However, **30** did not affect total STAT3 and -actin levels, confirming the inhibitory effects of carbazoles on STAT3 phosphorylation. Moreover, they analyzed the levels of phospho-STAT3 and lysine demethylase (LSD1) in the nuclear extract of HCC-2279 cells, at different concentrations of **30** (0, 1.25, 2.5, 5, 10, 20, 40 and 80 μM). ACB downregulated the nuclear pool of phospho-STAT3 pointing out that **30** suppresses the phosphorylation of STAT3, thereby translocating into the nucleus and reducing its DNA binding ability. LSD1 was exploited as nuclear marker and loading control for nuclear protein. The levels of LSD1, cytoplasmic STAT3 and *β*-actin did not vary.

These data were also confirmed by in silico studies that determined the acetyl group on the core carbazole as the most auspicious substituent. In particular, in different matched molecular series, compounds bearing the acetyl group showed the best cellular activities. An ethyl substituent is related to a modest solubility of the corresponding compounds, particularly in relation to chlorine substitutions.

On the contrary, they explored *N*-substituents and found the 3-(2-cyano-phenyl)-benzyl substitution as decoration with the highest effects in two out of three matched molecular series. Moreover, they extracted the SH2 domain from the crystal structure of the STAT3 homo-dimer and chose the pTyr as the center for in silico docking experiments. The 3-(2-cyano-phenyl)-benzyl substituted compounds such as **30**, **31**, and **32** demonstrated distinct features in molecular docking apart from conserved shape fit and hydrophobic interactions. The substituents on the carbazole moiety were buried in the pTyr binding site and thereby form hydrogen bonds via the acetyl and hydroxyethyl group. This provides a clarification of the better activity of the acetyl and ethyl decoration over the ethyl one, which misses the acceptor functionality resembling the pTyr in binding to the side chain of Arg609.

Diaz et al. in 2011 [118] explored the possible activity of Lestaurtinib against HL based on the knowledge that JAK/STAT pathway constitutive activation is involved in Hodgkin's lymphoma (HL) pathogenesis and the ability of Lestaurtinib (formerly known as CEP-701, 33 Figure 15) to inhibit JAK2 and FLT3 [119] in myeloproliferative disorders. Genomic gains of JAK2 [120], due to 9p24 gains [121], and SOCS1, a negative regulator of JAK/STAT signaling, are often somatically mutated and inactivated in HL [122]. Besides, constitutive activation of STAT3 has been documented in HL cell lines [123].

To verify, they treated five HL cell lines from refractory patients, L-428, L-1236, L-540, HDML-2, and HD-MY-Z with Lestaurtinib. They noticed a dose-dependent cell growth suppression (23%–66% at 300 nM) and an apoptotic increase (10%–64% at 300 nM) after treatment for 48 h. Furthermore, Lestaurtinib prevented JAK2, STAT5, and STAT3 phosphorylation and decreased the mRNA expression of its downstream antiapoptotic

target Bcl-xL. Furthermore, they determined after 1 h, phospho-JAK2 levels were reduced in all the HL cell lines by 46–94% at 300 nM, whereas no substantial changes were detected in JAK2 total protein expression.

**Figure 15.** Structure of lestaurtinib (CEP-70, **33**).

Lestaurtinib markedly hampered the phosphorylation of STAT5 and STAT3. However, no substantial changes in STAT5 and STAT3 total protein were assessed. Phospho-STAT5 and phosphoSTAT3 levels declined by 88–100% and by 97–100%, respectively, after 1 h of 300 nM of Lestaurtinib treatment. Bcl-xL, a pro-survival protein triggered by phosphory-lated STAT5 DNA binding, is upregulated in HL samples, and is implicated in apoptotic resistance in HRS cells. Reduced phosphorylation of STAT5 resulted in lowered mRNA expression of its downstream antiapoptotic effector Bcl-xL. Cells were treated for 1 h with Lestaurtinib at 300 nM discovering that Bcl-xL mRNA expression levels diminished by 52% in L-428, 28% in L-1236, 37% in L540, 55% in HDLM-2, and 71% in HD-MY-Z [124].

Subsequently, Santos et al. [125] carried out a phase 2 study of CEP-701 in which 22 JAK$^{2V617F}$-positive patients with myelofibrosis, for which few therapeutic options are available today, were treated with 80 mg, orally twice daily, of this drug. The total response rate for the patients was of 27%, with a median time to response of three months. Phosphorylated STAT3 levels decreased from baseline in responders while on therapy. A decline in spleen size occurred in three patients (50%). Two patients reached transfusion independency (9%). A dose decrease was necessary for 27% of patients. to reduce toxicity. Hematologic side effects such as anemia and thrombocytopenia were experienced by 14% and 23% of patients, respectively. Nonhematologic toxicity, including diarrhea (72%), nausea (50%), and vomiting (27%), were noted [125].

## 4. Conclusions

The academic community has extensively explored, over the years, heterocyclic compounds that represent more than half of the organic molecules known to exist. Among them a main role is played by carbazoles. This structural motif is found in different naturally occurring alkaloids. An ascending interest in these versatile compounds led both to the development of many synthetic strategies and their extraction from several plants. Indeed, they found an application both in science materials and the pharmaceutical field. In particular, some prior studies suggest that they could be promising anticancer agents. [126] According to some studies their effects could be due to the involvement of the JAK/STAT pathway. Signal transducers and activators of transcription (STATs) are a family of cytoplasmic proteins with roles like signal messengers and transcription factors engaged in normal cellular responses to cytokines and growth factors. Associated with STATs is the Janus kinases (JAK), belonging to the tyrosine kinases family, responsible for activating the

STAT cascade that initiates, ultimately, gene transcription. [127] Literature data report that carbazoles (Table 1) could act by downregulating STAT proteins (particularly STAT-3), and also affecting interleukins and i-NOS production.

**Table 1.** Carbazole derivatives as stat inhibitors.

| Compound | Name Compound | Structure Compound | References |
|---|---|---|---|
| **1** | Carbazole |  | Arbiser et al. [31] |
| **11** | 2-Hydroxycarbazole |  | Arbiser et al. [31] |
| **12** | Mahanine 3,11-dihydro-3,5-dimethyl-3-(4-methyl-3-pentenyl)-pyrano[3,2-*a*]carbazol-9-ol |  | Sarkar et al. [100] |
| **13a** | Dimethyl-5-(5-(6-methoxy-1,4-dimethyl-9*H*-carbazol-9-yl) pentyloxy)isophthalate |  | Saturnino et al. [32] |
| **13b** | Dimethyl-5-(6-(6-methoxy-1,4-dimethyl-9*H*-carbazol-9-yl) hexyloxy)isophthalate |  | Saturnino et al. [32] |
| **13c** | Dimethyl-5-(7-(6-methoxy-1,4-dimethyl-9*H*-carbazol-9-yl) heptyloxy)isophthalate |  | Saturnino et al. [32] |
| **14a** | Methyl-2-(5-(6-methoxy-1,4-dimethyl-9*H*-carbazol-9-yl) pentyloxy) benzoate |  | Saturnino et al. [32] |

**Table 1.** *Cont.*

| Compound | Name Compound | Structure Compound | References |
|---|---|---|---|
| **14b** | Methyl-2-(6-(6-methoxy-1,4-dimethyl-9*H*-carbazol-9-yl)hexyloxy)benzoate |  | Saturnino et al. [32] |
| **14c** | Methyl-2-(7-(6-methoxy-1,4-dimethyl-9*H*-carbazol-9-yl)heptyloxy)benzoate |  | Saturnino et al. [32] |
| **15** | 7-Hydroxy-1-methyl-9*H*-carbazol-2-yl-5-(dimethylamino)-naphthalene-1-sulfonate |  | Hou et al. [107]; Kang et al. [111] |
| **16** | EC-70124 (5*R*,7*S*,9*S*)-7,8-Dihydroxy-9-methyl-6,7,8,9-tetrahydro-5*H*,14*H*-17-oxa-4b,9a,15-triaza-5,9-methanodibenzo[b,h]cyclonona[jkl]cyclopenta[e]-as-indacene-14,16(15*H*)-dione |  | Cuenca-López et al. [113] Estupiñan et al. [114] |
| **17** | 1,4-Dimethyl-9*H*-carbazole |  | Botta et al. [115] |
| **18** | 6-Hydroxy-1,4-dimethyl-9*H*-carbazole |  | Botta et al. [115] |
| **19** | 6-Methoxy-1,4-dimethyl-9*H*-carbazole |  | Botta et al. [115] |
| **20** | Ethyl-5,8-dimethyl-9*H*-carbazole-3-carboxylate |  | Botta et al. [115] |

**Table 1.** *Cont.*

| Compound | Name Compound | Structure Compound | References |
|---|---|---|---|
| **21** | 6-Chloro-1,4-dimethyl-9*H*-carbazole | | Botta et al. [115] |
| **22** | 5,8-Dimethyl-9*H*-carbazole-3-sulfonamide | | Botta et al. [115] |
| **23** | 6-Methoxy-1,4-dimethyl-3-nitro-9*H*-carbazole | | Botta et al. [115] |
| **24** | Ethyl-5,8-dimethyl-9*H*-carbazole-3-nitro-carboxylate | | Botta et al. [115] |
| **25** | 6-Methoxy-1,4,9-trimethyl-carbazole | | Botta et al. [115] |
| **26** | 6-Methoxy-1,4-dimethyl-9-ethyl-carbazole | | Botta et al. [115] |
| **27** | 3-(3,4-Dicholophenyl)-6-(morpholine-4-carbonyl)-9*H*-carbazole-1-carboxamide | | Zimmermann et al. [116] |
| **28** | (*S*)-7-(2-(dimethylamino)-pyrrolidine-1-carbonyl)-3-(3-methoxyphenyl)-9*H*-carbazole-1-carboxamide | | Zimmermann et al. [116] |

**Table 1.** *Cont*.

| Compound | Name Compound | Structure Compound | References |
|---|---|---|---|
| **29** | 3-(1-Methyl-1*H*-indazol-5-yl)-7-(4-methylpiperazin-1-yl)-9*H*-carbazole-1-carboxamide |  | Zimmermann et al. [116] |
| **30** | 3′-((3-Acetyl-6-chloro-9*H*-carbazol-9-yl)methyl)-[1,1′-biphenyl]-2-carbonitrile |  | Baburajeev et al. [117] |
| **31** | 3′-((3-Chloro-6-ethyl-9*H*-carbazol-9-yl)methyl)-[1,1′-biphenyl]-2-carbonitrile |  | Baburajeev et al. [117] |
| **32** | 3′-((3-Chloro-6-(1-hydroxyethyl)-9*H*-carbazol-9-yl)methyl)-[1,1′-biphenyl]-2-carbonitrile |  | Baburajeev et al. [117] |
| **33** | CEP-701 Lestaurtinib (5*S*,6*S*,8*R*)-6-Hydroxy-6-(hydroxymethyl)-5-methyl-7,8,14,15-tetrahydro-5*H*-16-oxa-4b,8a,14-triaza-5,8-methanodibenzo[b,h]cycloocta[jkl]cyclopenta[e]-as-indacen-13(6*H*)-one |  | Diaz et al. [118] Shabbir et al. [119] Geyer et al. [124] Santos et al. [125] |

Figure 16 represents the mechanisms of action of the compounds cited in this review.

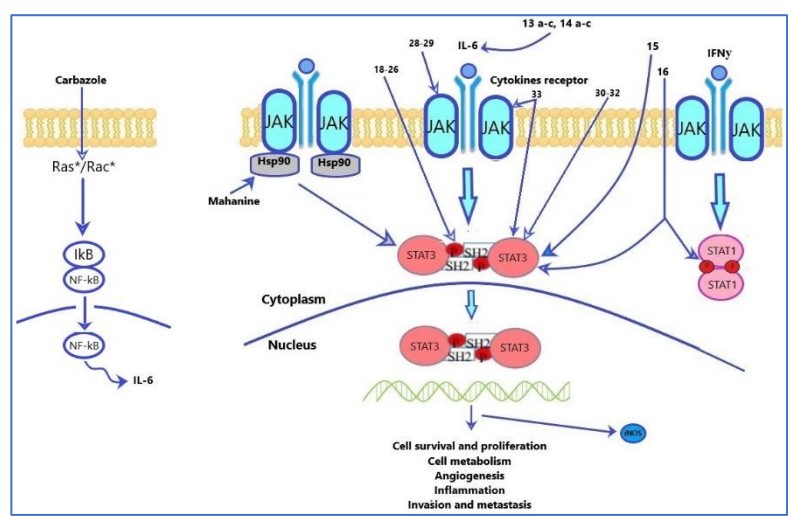

**Figure 16.** Mechanism of action of compounds carbazole mahanine and **13 a–c**, **14 a–c**, **15**, **16**, **18–26**, **28**, **29**, **30–33**.

Some of these studies have also been supported by in silico evaluations. It may be interesting to compare the activity between some successful carbazole and non-carbazole molecules targeting STATs. Indeed, HJC0123, cpd30, cpd188, and carbazolic compound 15 demonstrated their ability to target STAT3 and to exert antitumoral effects against breast cancer. Among them, HJC0123 exhibited the lower $IC_{50}$ values, amounting to 0.1–1.25 μM. Furthermore, it demonstrated to be orally bioavailable and suppressed tumoral growth in vivo [90]. Concerning compound **15**, its mechanism of action on STAT3 has been deepened, demonstrating its ability to inhibit phospho-STAT3 in the micromolar range. Moreover, other findings also reported interesting activity in an in vivo model of breast cancer in which the tumor suppression occurred after a few days of treatment. All this data, taken together, make compound **15** a promising carbazolic lead compound and a potential tool in the fight against cancer. However, further studies able to validate the importance of compound **15** and of other promising carbazoles for clinical use must be carried out. If they succeed, carbazoles could represent a valid alternative to conventional treatments, allowing us to overcome the phenomena of drug resistance, entering, in this way, into the plethora of target therapies.

**Author Contributions:** Conceptualization, A.C. (Anna Caruso) and C.S.; methodology, A.B.; software, G.S.; resources, A.C. (Anna Caruso) and A.B.; data curation, A.C. (Alessia Carocci); writing—original draft preparation, A.C. (Anna Caruso) and A.B.; writing—review and editing, A.C. (Alessia Carocci); supervision, M.S.S. and C.S. All authors have read and agreed to the published version of the manuscript.

**Funding:** This research received no external funding.

**Institutional Review Board Statement:** Not applicable.

**Informed Consent Statement:** Not applicable.

**Data Availability Statement:** Not applicable.

**Conflicts of Interest:** The authors declare no conflict of interest.

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
