# Peer review of "Carbazole Derivatives as STAT Inhibitors: An Overview"

_applsci, doi:10.3390/app11136192_

Round 1

Reviewer 1 Report

I enclose below my suggestions for authors

Author Response

Dear Editor,

We would like to thank the reviewers for the useful suggestions regarding our manuscript and for the opportunity to submit a revised version in response to their criticisms. The comments and suggestions of the reviewers have been very helpful, and, accordingly, we have implemented the manuscript. Please find below a detailed point-by-point description of our responses to each of the reviewer’s comments.

Revisore 1

In the review entitled "Recent advances in the developments of carbazole derivatives as STAT inhibitors: a review" Anna Caruso and colleagues discuss recent literature concerning the significance of carbazoles as STAT inhibitors in still uncurable, serious diseases. At first, authors describe the role of JAK-STAT signaling pathway in the development highly malignant cancers and psoriasis. Next, they underline the impact on currently investigated and tested STATs inhibitors as antiproliferative agents. Finally, they make a comprehensive analysis of the potential mechanism of action of carbazoles as STATs inhibitors in highly aggressive cancer cells. The review include a balanced, comprehensive and critical view of the very important research area. However it needs further corrections.

The introduction contains too much unnecessary contents regarding the general and known information about other heterocycles instead of focus mostly on carbazoles only. Done

In some places in the text I found the sentence: “Error! Reference source not found” that cannot exist in final version.

We have reviewed the references carefully in the text

A large number language and stylistic errors make the text difficult to understand.  

Thanks for the suggestion, the stylistic errors has been corrected

I would suggest as follows: - the introduction should concentrate only on carbazoles, especially the ones that are STATs inhibitors –  Done

in the conclusion I would suggest to compare the most active STATs carbazole inhibitors with others non-carbazole derivatives most active STATs inhibitors to highlight the potential role of the first one as antiproliferative agents Done

Language: The layout most of the sentences is hardly legible, what makes this excellent review unintelligible

Thanks for the suggestion, the language has been simplified

After proofreading, I recommend the manuscript for publication.

Reviewer 2 Report

Review of the paper: “RECENT ADVANCES IN THE DEVELOPMENTS OF CARBAZOLE DERIVATIVES AS STAT INHIBITORS: A REVIEW.”

The work to be evaluated is a current and high-impact topic, however, it is not adequately covered in this paper.

For a review, most of the references cited in the paper are not recent (although the title indicates so). It is remarkable that similar reviews published in more recent years are not cited, at least in the Introduction. In addition, I have noticed that the number of self-citations in this part is very high.

I found many typographical errors, but that is not so important.

However, the reading has been challenging because the citation corresponding to what was exposed was far away, or simply unrelated to it. For example:

"Knolker et al. thoroughly reviewed the synthesis of these carbazole alkaloids [26]". And ref 26 is "Roy, J.; Jana, A. K.; Mal, D., Recent trends in the synthesis of carbazoles: an update. Tetrahedron 2012, 68 (31), 6099-6121".

"Hou et al have previously been fascinated by the possibility of targeting STAT3 pathways [123,124]."  However “Hou et al” is reference 128. References 123, 124 are unrelated.

"Arbiser et al in 2006 fractionated coal tar through..." or "In 2012, Sarkar et al." there are no references for these comments.

Some of the references are written twice with different numbers, such as ref 34 and 108 or Reference 120 and 35.

The abbreviations in the journal name need to be rewritten. Most of them have the long name.

In my opinion, the selection of references to be a review of the subject has not been good, sometimes they are too specific or not related to that they describe.

Finally, Cpd30, Cpd188, OBP31121, HO-3867, LLL12 are only named but their structures are not shown in the text. Perhaps, a scheme with some of them will help to better understand all the information presented. Finally, figure 8 should introduce 3a-c (n=5-7) to avoid confusion.

Therefore, although the idea is interesting and up-to-date, the presentation could be much improved. A review paper that is supposed to would be a helpful to many researchers cannot have so many inaccuracies in the citations described.

That’s my opinion although the final decision depends on the editor.

Author Response

 We would like to thank the reviewer for the useful suggestions regarding our manuscript and for the opportunity to submit a revised version in response to their criticisms. The comments and suggestions of the reviewer have been very helpful, and, accordingly, we have implemented the manuscript. Please find below a detailed point-by-point description of our responses to each of the reviewer’s comments.

Review of the paper: “RECENT ADVANCES IN THE DEVELOPMENTS OF CARBAZOLE DERIVATIVES AS STAT INHIBITORS: A REVIEW.”

The work to be evaluated is a current and high-impact topic, however, it is not adequately covered in this paper.

For a review, most of the references cited in the paper are not recent (although the title indicates so). It is remarkable that similar reviews published in more recent years are not cited, at least in the Introduction. In addition, I have noticed that the number of self-citations in this part is very high.

References have been updated with more recent publications, such as:

  1. Bhambhani S.; Kondhare K. R.; Giri A.P.; Diversity in Chemical Structures and Biological Properties of Plant Alkaloids. Molecules 2021, 26, 3374.
  2. Gao P., Niu N.,  Wei T., Tozawa H., Chen X.,  Zhang C., Zhang J.,  Wada Y., Kapron C. M.,  Liu J. The roles of signal transducer and activator of transcription factor 3 in tumor angiogenesis. Oncotarget. 2017; 8, 69139-69161
  3. Miklossy G., Hilliard T.S., Turkson J. Therapeutic modulators of STAT signalling for human diseases. Nat Rev Drug Discov. 2013; 12(8): 611–629.
  4. Zhang N, Zheng Q, Wang Y, Lin J, Wang H, Liu R, Yan M, Chen X, Yang J, Chen X. Renoprotective Effect of the Recombinant Anti-IL-6R Fusion Proteins by Inhibiting JAK2/STAT3 Signaling Pathway in Diabetic Nephropathy. Front Pharmacol. 2021, 12, 681424.
  5. Gharibi T., Babaloo Z., Hosseini A, Abdollahpour-alitappeh M., Hashemi V., Marofi F, Nejati K., Baradaran B.. Targeting STAT3 in cancer and autoimmune diseases. Eur. J. Pharmacol. 2020, 878, 173107.
  6. Issa S., Prandina A., Bedel N., Rongved P., Yous S., Le Brogne M., Bouaziz Z. Carbazole scaffolds in cancer therapy: a review from 2012 to 2018. J Enzyme Inhib Med Chem 2019; 34(1); 1321-1346

119 Tibes R.; Bogenberger J.M.; Benson K.L.; Mesa R. A. Current outlook on molecular pathogenesis and treatment of myeloproliferative neoplasm. Mol Diagn Ther 2012; 16; 269-283.

Some self-citations have been eliminated

I found many typographical errors, but that is not so important.

These typographical errors has been corrected

However, the reading has been challenging because the citation corresponding to what was exposed was far away, or simply unrelated to it. For example:

"Knolker et al. thoroughly reviewed the synthesis of these carbazole alkaloids [26]". And ref 26 is "Roy, J.; Jana, A. K.; Mal, D., Recent trends in the synthesis of carbazoles: an update. Tetrahedron 2012, 68 (31), 6099-6121".

"Hou et al have previously been fascinated by the possibility of targeting STAT3 pathways [123,124]."  However “Hou et al” is reference 128. References 123, 124 are unrelated.

"Arbiser et al in 2006 fractionated coal tar through..." or "In 2012, Sarkar et al." there are no references for these comments.

This has been corrected

Some of the references are written twice with different numbers, such as ref 34 and 108 or Reference 120 and 35. Done

The abbreviations in the journal name need to be rewritten. Most of them have the long name. Done

In my opinion, the selection of references to be a review of the subject has not been good, sometimes they are too specific or not related to that they describe.

The references have all been revised and some of these, too specific, have been eliminated

Finally, Cpd30, Cpd188, OBP31121, HO-3867, LLL12 are only named but their structures are not shown in the text. Perhaps, a scheme with some of them will help to better understand all the information presented. Done, see figure 5

Finally, figure 8 should introduce 3a-c (n=5-7) to avoid confusion. Done

Round 2

Reviewer 1 Report

Please find attached my comments and suggestions in the file below

Author Response

At the beginning I would like to thank to A. Caruso and colleagues for making a partial corrections in their article according my first review. Regarding the language, the chapters entitled introduction and ‘STAT proteins: novel molecular targets for cancer drug discovery’ are generally written correctly what makes an article much better understandable. However some sentences need some minor revisions. Nevertheless, the further chapters still need major corrections to make them clear for readers. I would suggest to engage the native speaker to solve this problem. Concerning the content of some chapters, I propose make changes according their text headers.

The paper was reviewed by a native speaker and the content of some chapters as been changed .

Namely, in the chapter entitled ‘STAT proteins: novel molecular targets for cancer drug discovery’ I suggest to focus only on the significance of mammalian STATs in the development or apoptosis of certain cancers. Description of their influence on the activity of immune system will complete this chapter.

Thanks, we have modified the chapter following the suggestion.

Next, in a chapter entitled ‘STATs inhibitors’ I miss the miss the division of these inhibitors for natural and synthetic. Next, the description almost of all of them is not in order. I advice to that in all chapters in a wellorganized and controlled way.

The chapter ‘STATs inhibitors’  has divided in paragraphs: “Peptides and small molecules as STATs inhibitors”; “Natural compounds as STATs inhibitors” and “Carbazoles as STAT inhibitors”

Finally, I see again twice in the text, the sentence ‘Error! Reference source not found’.

The sentence "Errore! Fonte di riferimento non trovata” has been removed twice. We apologize for the inconvenience.

Please remove it. Please find below, the suggestions of some sentences:

  1. In the text: Many carbazoles derivated from plants are provided of biological activities, comprising antitumor, psychotropic, anti-inflammatory, antihistaminic, antibiotic, and antioxidant

My suggestion: Number of carbazoles derived from plants possess comprising antitumor, psychotropic, anti-inflammatory, antihistaminic, antibiotic and antioxidant activities  DONE

  1. In the text: Concerning the biological activity, many research groups assessed the antiproliferative action of carbazoles on different types of tumoral cells such as breast, cervical, ovarian, hepatic, oral cavity, and small-cell lung cancer

My suggestion: Concerning the biological activity, many research groups assessed the antiproliferative action of carbazoles against different types of tumor cells such as breast, cervical, ovarian, hepatic, oral cavity, and small-cell lung cancer  DONE

  1. In the text: Signal Transducers and Activators of Transcription (STATs) constitute a group of cytoplasmic proteins functioning as signal messengers and transcription factors engaaged in the usual cellular reactions to cytokines and growth factors.

My suggestion: Signal Transducers and Activators of Transcription (STATs) constitute a group of cytoplasmic proteins that function as signal messengers and transcription factors engaged in critical cellular events with use of multiple cytokines and growth factors. DONE

  1. In the text: Tyrosine kinases involved in STAT activation include Janus kinase (JAK) and Src kinase families. Since tyrosine has been phosphorylated, two STAT monomers develop dimers via mutual phosphotyrosine-SH2 interactions, migrate to the nucleus, and bind to STAT-specific DNA-response elements of target genes causing gene transcription.

My suggestion: Since tyrosine has been phosphorylated, two STAT monomers develop dimers via mutual phosphotyrosine-SH2 interactions, migrate to the nucleus and bind to STAT-specific DNA-response elements of target genes what cause gene transcription. DONE

  1. In the text: Interferon (IFN) stimulation triggers STAT1 signalling and sustains immune system by balancing both the growth and apoptosis of immune cells [32]. Indeed, STAT1 shortcoming reverses IFN reactivity, which make mice more susceptible to bacterial and viral infections to the death [33]. Moreover, the lack of responsiveness increases tumour formation, indicating that STAT1 possess a tumoursuppressive function; even though up-to-date findings suggest that STAT1 is involved in carcinogenesis at a more comlex level [30]. STAT1 signalling governs T helper type 1 (TH1) cell-specific cytokine production that changes both immune functions and inflammatory responses by modifying the ratio between TH1 and TH2 cells [32] DONE

My suggestion: STAT1 signaling suppresses type 17 immunity what makes mice more susceptible both to bacterial and viral infections. Additionally, STAT1 exerts multidirectional antitumor effect. It blocks the cell cycle progression and angiogenesis and induces proapoptotic genes. Moreover, STAT1 signalling governs T helper type 1 (TH1) cell-specific cytokine production that changes both immune functions and inflammatory responses by modifying the ratio between TH1 and TH2 cells. DONE

Reviewer 2 Report

Evaluation of the paper: “RECENT ADVANCES IN THE DEVELOPMENTS OF CARBAZOLE DERIVATIVES AS STAT INHIBITORS: A REVIEW.”

The submitted work shows slight improvements over the previous version, but still contains some deficiencies that have not been corrected. 

I listed some of them by my opinion:

1).- Reference 1 is too specific in relation to the information given in the first paragraph.

2).-The text say “at first isolated by Graebe and Glazer in 1872,” any reference for this information is given (C. Graebe, C. Glaser, Justus Liebigs Annalen der Chemie, 1872, 163, 343-30)

3).-Same comment for “In 1965, Chakraborty et al. reported the isolation”. No reference for this information (DP Chakraborty, BK Barman, PK Bese, Tetrahedron, 1965, 21, 681).

4).- murrayanine (2) from Murraya koenigii Spreng [9,10]. References 9 and 10 are not related. It could be placed after “encouraging biological effects displayed by several carbazole alkaloids.”

5).-The text says:” Indeed, their structural motif is predominant in different artificial materials and naturally occurring alkaloids.” Reference 1 could be located here and my opinion ref 12.

6).- The text says: ”Many carbazoles derivated from plants are provided of biological activities, comprising antitumor, psychotropic, anti-inflammatory, antihistaminic, antibiotic, and antioxidant [12].”  But Reference 12 is: “Synthesis and evaluation of cytotoxic activities of new guanidines derived from carbazoles. Bioorg. Med. Chem. Lett. 2014, 24 (2), 467-472”. Perphaps another reference should be cited here in reference with the text.

7).- The text says:” Knolker et al. reviewed thoroughly the preparation methods of these carbazole alkaloids [20].”  In the last revision I commented: “"Knolker et al. thoroughly reviewed the synthesis of these carbazole alkaloids [26]". And ref 26 is "Roy, J.; Jana, A. K.; Mal, D., Recent trends in the synthesis of carbazoles: an update. Tetrahedron 2012, 68 (31), 6099-6121".  Surprinsigly, the only thing that have changed in this new version is the number of the reference, because now reference 20 is still "Roy, J.; Jana, A. K.; Mal, D., Recent trends in the synthesis of carbazoles: an update. Tetrahedron 2012, 68 (31), 6099-6121". Knolker’s reference is number 11.

8).- The text says: ”Due to the growing interest in carbazoles bioactivities, different synthetic strategies have been set up and reported in the literature [19].”. Reference 20 could be here as well.

9), “(11, Figure 7) could downregulate stat3-mediated transcription.” In the rest of the text STAT3 is written in capital letters.

10).- “carcinogenesis at a more comlex level [30].” Comlex should be changed for complex.

12) .- “STAT5A and STAT5B [43].” A and B in capital letters?

13).- “For this reason, this research group focused their attention on carbazole (1) included in coal tar.” Number 1 is not in bracket.

14).- Again, the text says “Hou et al. have been previously fascinated by the possibility of targeting STAT3 pathways [97,98]. And reference 97 and 98 are from a different author. Hou’s reference is 102.

15) Something has caught my attention and could indicate the authors' interest in correcting this paper. In the previous version, the number of references was 143 and in the new version it is 119, but in Table 1, which summarises the different carbazole derivatives uses as statin inhibitors, the number of references that appear there are the same as in the old version (from 120 to 143).

I have carefully read the article several times and although the title is "Recent Advances in the Development of Carbazole Derivatives as Stat Inhibitors: A Review", I still do not see the recent advances reflected, so perhaps the problem is the chosen title which might not be the right one. I suggest something similar to “ Carbazole derivatives as Stat Inhibitors: A Overview”

Therefore, although the idea is interesting and up-to-date, the presentation could be much improved. A review paper that is supposed to would be a helpful to many researchers cannot have so many inaccuracies in the citations described.

That’s my opinion although the final decision depends on the editor.

Author Response

The submitted work shows slight improvements over the previous version, but still contains some deficiencies that have not been corrected. 

I listed some of them by my opinion:

1).- Reference 1 is too specific in relation to the information given in the first paragraph. DONE

2).-The text say “at first isolated by Graebe and Glazer in 1872,” any reference for this information is given (C. Graebe, C. Glaser, Justus Liebigs Annalen der Chemie, 1872, 163, 343-30) DONE

3).-Same comment for “In 1965, Chakraborty et al. reported the isolation”. No reference for this information (DP Chakraborty, BK Barman, PK Bese, Tetrahedron, 1965, 21, 681). DONE

4).- murrayanine (2) from Murraya koenigii Spreng [9,10]. References 9 and 10 are not related. It could be placed after “encouraging biological effects displayed by several carbazole alkaloids.” DONE

5).-The text says:” Indeed, their structural motif is predominant in different artificial materials and naturally occurring alkaloids.” Reference 1 could be located here and my opinion ref 12. DONE

6).- The text says: ”Many carbazoles derivated from plants are provided of biological activities, comprising antitumor, psychotropic, anti-inflammatory, antihistaminic, antibiotic, and antioxidant [12].”  But Reference 12 is: “Synthesis and evaluation of cytotoxic activities of new guanidines derived from carbazoles. Bioorg. Med. Chem. Lett. 2014, 24 (2), 467-472”. Perphaps another reference should be cited here in reference with the text. DONE

7).- The text says:” Knolker et al. reviewed thoroughly the preparation methods of these carbazole alkaloids [20].”  In the last revision I commented: “"Knolker et al. thoroughly reviewed the synthesis of these carbazole alkaloids [26]". And ref 26 is "Roy, J.; Jana, A. K.; Mal, D., Recent trends in the synthesis of carbazoles: an update. Tetrahedron 2012, 68 (31), 6099-6121".  Surprinsigly, the only thing that have changed in this new version is the number of the reference, because now reference 20 is still "Roy, J.; Jana, A. K.; Mal, D., Recent trends in the synthesis of carbazoles: an update. Tetrahedron 2012, 68 (31), 6099-6121". Knolker’s reference is number 11. DONE

8).- The text says: ”Due to the growing interest in carbazoles bioactivities, different synthetic strategies have been set up and reported in the literature [19].”. Reference 20 could be here as well. DONE

9), “(11, Figure 7) could downregulate stat3-mediated transcription.” In the rest of the text STAT3 is written in capital letters. Done

10).- “carcinogenesis at a more comlex level [30].” Comlex should be changed for complex.

Thanks for the suggestion, but following the advice of the other reviewer, this has been deleted from the text.

12) .- “STAT5A and STAT5B [43].” A and B in capital letters?

“STAT5A and STAT5B”  has been changed to “Stat5a and Stat5b” as reported by Lin et al. reference 50.

13).- “For this reason, this research group focused their attention on carbazole (1) included in coal tar.” Number 1 is not in bracket. Done

14).- Again, the text says “Hou et al. have been previously fascinated by the possibility of targeting STAT3 pathways [97,98]. And reference 97 and 98 are from a different author. Hou’s reference is 102. Done

15) Something has caught my attention and could indicate the authors' interest in correcting this paper. In the previous version, the number of references was 143 and in the new version it is 119, but in Table 1, which summarises the different carbazole derivatives uses as statin inhibitors, the number of references that appear there are the same as in the old version (from 120 to 143).

Thanks for the suggestion, in table 1 the number of references has been corrected.

I have carefully read the article several times and although the title is "Recent Advances in the Development of Carbazole Derivatives as Stat Inhibitors: A Review", I still do not see the recent advances reflected, so perhaps the problem is the chosen title which might not be the right one. I suggest something similar to “ Carbazole derivatives as Stat Inhibitors: A Overview” DONE

Therefore, although the idea is interesting and up-to-date, the presentation could be much improved. A review paper that is supposed to would be a helpful to many researchers cannot have so many inaccuracies in the citations described.

Thanks, we apologize for the inconvenience. The citations have been revised carefully and corrected.

Round 3

Reviewer 1 Report

At the beginning I would like to thank again the authors for the further corrections that made their article easy to read and understand. However minor language corrections are still necessary. Please take a look carefully the chapters: STAT proteins: novel molecular targets for cancer drug discovery and STAT inhibitors. In the chapter entitled STAT inhibitors highlight the the certain antiproliferative/antinflammatory activity of each described there STAT inhibitor.

Author Response

At the beginning I would like to thank again the authors for the further corrections that made their article easy to read and understand. However minor language corrections are still necessary. Please take a look carefully the chapters: STAT proteins: novel molecular targets for cancer drug discovery and STAT inhibitors.

DONE

In the chapter entitled STAT inhibitors highlight the the certain antiproliferative/antinflammatory activity of each described there STAT inhibitor.

The certain antiproliferative / anti-inflammatory activity of each described there STAT inhibitor has been highlight.

Reviewer 2 Report

Review of the paper: “CARBAZOLE DERIVATIVES AS STAT INHIBITORS: AN OVERVIEW.”

I have only a few comments which I list below:

Page 2.- “Many natural carbazoles and their synthetic derivatives contain annulated rings, as ellipticine (4), carbazomycin B (5), carbazomadurin A (6), clausenamine A (7), staurosporine (8) [21]”. This phrase could be more accurate because carbazole is already a tricyclic compound, so only compounds 4 and 8 contain a extra ring, but compounds 5 and 6 just have more substituents, and compound 7 is a dimer.

Page 17.- “Subsequently, Santos et al. [125] carried out a phase 2 study of CEP-701 in which patients were treated with 80 mg/BID. The total response rate was 27% with a median time to response of 3 months. Just in three patients a decline in spleen size occurred (50%). Two patients reached transfusion independency (9%). A dose decrease was necessary for 27% of patients. to reduce toxicity. Hematologic side effects as anemia and thrombocytopenia experienced at 14 and 23%, respectively. Nonhematologic toxicity included diarrhea (72%), nausea (50%), and vomiting (27%) were noticed [125].“ In my opinion, this paragraph is a bit unclear, it should be checked with the original paper for better writing.

Page 7.- HO-3867 (Figure 5), “curcuumin” analog, not many but some minor mistakes like this it is posible to find in the text.

Page 15.- b-actin and beta-actin (in greek letter) is found in the same page.

Check format of the references, for example ref 35.

I consider that the article could be recommended for publication in this journal when several minor errors have been corrected. That is my opinion, although the final decision is up to the editor.

Author Response

I have only a few comments which I list below:

Page 2.- “Many natural carbazoles and their synthetic derivatives contain annulated rings, as ellipticine (4), carbazomycin B (5), carbazomadurin A (6), clausenamine A (7), staurosporine (8) [21]”. This phrase could be more accurate because carbazole is already a tricyclic compound, so only compounds 4 and 8 contain a extra ring, but compounds 5 and 6 just have more substituents, and compound 7 is a dimer.

This phrase has been changed to “Many natural carbazoles and their synthetic derivatives contain different substituents on the carbazole ring as carbazomycin B (4), carbazomadurin A (5), others present a quadricyclic or pentacyclic structure, as ellipticine (6) and staurosporine (7) respettivelly, or can appear as dimers, for example clausenamine A (8)”

Page 17.- “Subsequently, Santos et al. [125] carried out a phase 2 study of CEP-701 in which patients were treated with 80 mg/BID. The total response rate was 27% with a median time to response of 3 months. Just in three patients a decline in spleen size occurred (50%). Two patients reached transfusion independency (9%). A dose decrease was necessary for 27% of patients. to reduce toxicity. Hematologic side effects as anemia and thrombocytopenia experienced at 14 and 23%, respectively. Nonhematologic toxicity included diarrhea (72%), nausea (50%), and vomiting (27%) were noticed [125].“ In my opinion, this paragraph is a bit unclear, it should be checked with the original paper for better writing. DONE

Page 7.- HO-3867 (Figure 5), “curcuumin” analog, not many but some minor mistakes like this it is posible to find in the text. DONE

Page 15.- b-actin and beta-actin (in greek letter) is found in the same page. DONE

Check format of the references, for example ref 35. DONE

I consider that the article could be recommended for publication in this journal when several minor errors have been corrected. That is my opinion, although the final decision is up to the editor.